# Predicting Disability Using a Nomogram of the Tilburg Frailty Indicator (TFI)

**DOI:** 10.3390/healthcare11081150

**Published:** 2023-04-17

**Authors:** Robbert J. Gobbens, Livia M. Santiago, Izabella Uchmanowicz, Tjeerd van der Ploeg

**Affiliations:** 1Faculty of Health, Sports and Social Work, Inholland University of Applied Sciences, 1081 HV Amsterdam, The Netherlands; 2Zonnehuisgroep Amstelland, 1186 AA Amstelveen, The Netherlands; 3Department Family Medicine and Population Health, Faculty of Medicine and Health Sciences, University of Antwerp, 2610 Antwerp, Belgium; 4Department of Tranzo Academic Centre for Transformation in Care and Welfare, Faculty of Behavioural and Social Sciences, Tilburg University, 5037 AB Tilburg, The Netherlands; 5Faculty of Medicine, Federal University of Rio de Janeiro, Rio de Janeiro 21941-912, Brazil; 6Department of Nursing and Obstetrics, Faculty of Health Sciences, Wroclaw Medical University, 50-367 Wroclaw, Poland; 7Institute of Heart Diseases, University Hospital, 50-566 Wroclaw, Poland

**Keywords:** disability, frailty, older people, nomogram, Tilburg Frailty Indicator, Groningen Activity Restriction Scale, difficulty in walking, unexplained weight loss

## Abstract

Disability is associated with lower quality of life and premature death in older people. Therefore, prevention and intervention targeting older people living with a disability is important. Frailty can be considered a major predictor of disability. In this study, we aimed to develop nomograms with items of the Tilburg Frailty Indicator (TFI) as predictors by using cross-sectional and longitudinal data (follow-up of five and nine years), focusing on the prediction of total disability, disability in activities of daily living (ADL), and disability in instrumental activities of daily living (IADL). At baseline, 479 Dutch community-dwelling people aged ≥75 years participated. They completed a questionnaire that included the TFI and the Groningen Activity Restriction Scale to assess the three disability variables. We showed that the TFI items scored different points, especially over time. Therefore, not every item was equally important in predicting disability. ‘Difficulty in walking’ and ‘unexplained weight loss’ appeared to be important predictors of disability. Healthcare professionals need to focus on these two items to prevent disability. We also conclude that the points given to frailty items differed between total, ADL, and IADL disability and also differed regarding years of follow-up. Creating one monogram that does justice to this seems impossible.

## 1. Introduction

Population aging is occurring all over the world and is accompanied by an increasing number of people with disability; after all, the prevalence of disability is associated with aging. In the USA, the prevalence of disability is 10.6% among people aged 18–64 years and 35.2% for those aged 65 years or older [1]. There are different definitions of disability. According to the World Health Organization (WHO), disability has three dimensions referring to impairment in a person’s body structure or function (e.g., loss of a limb), or mental functioning (e.g., loss of memory), activity limitation (e.g., difficulty in walking), and participation restrictions in performing daily activities (e.g., engaging in recreational and social activities) [2]. In research focusing on older people, disability is often defined as having difficulty conducting activities of daily living (ADL) and/or instrumental activities of daily living (IADL) [3,4]; the second dimension was established by the WHO (activity limitation). Examples of ADL include getting on and off the toilet and standing up from sitting in a chair; doing the shopping, washing, and ironing clothes refer to IADL [5,6]. IADL disability reflects a less severe form of disability than ADL disability [7,8]. In a sample of Polish people aged ≥60 years, it was observed that 35.75% and 17.13% reported at least one problem with IADL and ADL, respectively [9]. In a sample of Dutch people aged ≥75 years, these percentages were 54.6% for IADL disability and 67.4% for ADL disability [10].

Disability is associated with adverse outcomes in older people, such as increased use of health care [11,12], lower quality of life [10,13], and premature death [14,15,16]. Early identification, prevention, and intervention targeting older people living with a disability is, therefore, of utmost importance. To achieve this, knowledge of factors influencing disability is essential. In Poland, it was shown that the occurrence of disability is influenced by a lack of social contacts, multimorbidity, pain, and the presence of barriers in the environment for older people [9]. In the Netherlands, risk groups include those living alone, women, people who are widowed or divorced, and those with a low educational level [17]. In addition to these risk factors and at-risk groups, frailty can be considered a major predictor of disability [18,19,20]. A systematic review using 28 studies showed that physical frailty indicators could predict ADL disability; the most powerful predictors were slow gait speed and low physical activity [20]. A more recent systematic review and meta-analysis including 20 studies quantitatively showed that frail older people were more likely to develop or have more severe ADL and IADL disability [19]. Finally, a systematic review and meta-analysis of prospective cohort studies including 32,998 people aged 60 years or older also concluded that those who are frail have the highest relative risk of disability [18]. 

As with disability, there are also different definitions of frailty [21]. There are definitions that view frailty exclusively as a concept that refers only to physical limitations that older people may have. One measurement tool that goes along with these definitions is the phenotype of frailty frequently used in studies [22]. This tool contains five physical criteria by which a healthcare professional can determine if a person is frail: weakness, unintentional weight loss, poor endurance, slowness, and low physical activity [22]. As a counterpart to physical frailty definitions, there are also definitions that emphasize the multidimensional nature of frailty. The measurement tools that align with these definitions not only pay attention to physical limitations but also to the psychological and social limitations of older people. Examples of such tools are the Frailty Index [23], based on the Canadian Study of Health and Aging (CSHA) Cumulative Deficit Model, and the Tilburg Frailty Indicator (TFI) [24]. 

As mentioned, many studies have been performed that aimed at examining the association between frailty and disability and, in particular, predicting disability by frailty [18,19,20]. In general, in these studies, frailty was assessed using a tool containing physical items (e.g., the phenotype of frailty). Studies using a multidimensional tool are rarer. In the present study, we used the TFI items belonging to physical, psychological, and social frailty as predictors of disability. We have assigned one point per item if there was a deficit. Linear and logistic regressions are frequently used as techniques to develop a prediction model; however, the relationship between TFI items and disability may differ. In this study, we aimed to develop nomograms with the TFI items as predictors by using cross-sectional data and longitudinal data, focusing on the prediction of disability, where we distinguish between total disability, ADL disability, and IADL disability. We will show that the nomograms differ per outcome and per time point. The different time points refer to the longitudinal aspect of the study. The knowledge we gather and present in the nomograms can support healthcare professionals in determining the risk of disability in community-dwelling older people. 

## 2. Methods

### 2.1. Study Population and Data Collection

For the present study, we followed the guidelines for reporting observational studies (STROBE) [25]. We used a sample of 479 community-dwelling people aged ≥75 years, referring to a 42% response rate, which has also been used in previous studies [24,26]. All participants were residents of Roosendaal, a municipality with 78,000 inhabitants in the south of the Netherlands. In June 2008 (T0), they received a self-report questionnaire including measures for assessing frailty and disability. A subset of all participants completed the same questionnaire five years later, in June 2013 (T5) (*n* = 160), and once more, four years later, in June 2017 (T9) (*n* = 77). 

### 2.2. Measures

#### 2.2.1. Groningen Activity Restriction Scale (GARS)

We used a continuous disability scale, the Groningen Activity Restriction Scale (GARS), for assessing disability (total) [27]. This self-report questionnaire contains 18 items; each item has 4 response categories (1 = yes, I can do that easily and without help; 2 = yes, I can do that without help, but it takes some effort; 3 = yes, I can do that without help but it takes a lot of effort; 4 = no, I cannot do that without help). The disability total score ranges from 18 (no disability) to 72 (maximum disability). A cut-off point of 29 has been established [28]; therefore, people with scores of 29 or higher can be considered people with disability. The two continuous subscales of the GARS were used for assessing ADL disability and IADL disability. The ADL subscale consists of 11 items, and the IADL consists of 7 items, with scores ranging from 11 to 44 and 7 to 28, respectively; higher scores indicate greater disability. Previous studies have shown that the GARS has good psychometric properties (reliability, validity) for assessing disability in older people [5,6].

#### 2.2.2. Frailty

We used the items from part B of the TFI as predictor variables. This part contains 15 items; 11 items have two response categories (yes, no), and 4 items have three response categories (yes, sometimes, no), dichotomized to yes, no. Eight items belong to physical frailty: physically unhealthy, unexplained weight loss, difficulty in walking, difficulty in maintaining balance, poor hearing, poor vision, lack of strength in the hands, and physical tiredness. Four items refer to psychological frailty: problems with memory, feeling down, feeling nervous or anxious, and unable to cope with problems. Finally, three items indicate social frailty: living alone, lack of social relations, and lack of social support [24]. The TFI total score ranges from 0 to 15; the higher the score, the more frailty is present in an individual. Many studies have shown that the TFI has good psychometric properties, evidenced by good reliability (internal consistency, test–retest reliability) and validity (criterion, construct) [29]. 

### 2.3. Statistical Analysis

We used counts and percentages to describe the baseline characteristics of the participants. For the multivariate analysis, we used linear regression with all 15 predictor variables (15 items of the TFI) at each time point. The predictive performance of the models was measured using the R-square statistic (*Rsq*). Values of *Rsq* toward 1 indicate good model performance. Nomograms were constructed based on the transformed coefficients (coefficients divided by the maximum of the coefficients and multiplied by 100) of the linear regression model. For all analyses, we used R version 3.4.4. [30].

### 2.4. Ethical Considerations

For this study, medical ethics approval was not necessary as particular treatments or interventions were not offered or withheld from respondents. The integrity of the respondents was not encroached upon as a consequence of participating in this study, which is the main criterion in medical–ethical procedures in the Netherlands [31]. This study was conducted according to the guidelines for good clinical practice. The researchers did not make the questionnaire long, so the burden on participants was limited; the average time for completing the questionnaire was 20 min. In addition, the questionnaire contained measures (GARS, TFI) that have already been used in many previous studies among older people.

## 3. Results

### 3.1. Characteristics of the Participants

Table 1 presents the characteristics of the participants at T0, T5, and T9. At baseline, the sample consisted of 272 women (56.8%), 49.8% of the participants were married or cohabiting, and most participants had secondary education (46.5%). The group of participants decreased over time as expected (*n* = 479, *n* = 160, and *n* = 77, respectively). The mean values of the frailty scores and the disability scores increased over time.

The change in distribution of the categorial variables over time was tested using the chi-square test. The *p*-values are shown in Table 2. The change in monthly income from T0 to T5 stands out in Table 2 (*p*-value < 0.001). 

The change in distribution of the continuous variables over time was tested using the paired Wilcoxon test. For all comparisons, the *p*-values were <0.001 (not surprisingly for the variable age). Table 3 shows the characteristics of the variables in the B-part of the TFI (i.e., our predictor variables) at T0. For all predictors, except ‘lack of social relations’, the ‘yes’ category had the lowest frequency.

### 3.2. Prediction of Total Disability

Table 4 shows the points for the items in the nomograms and the performance of the underlying models (*Rsq*-values) over time in predicting the total disability scores. Table 4 shows stable *Rsq*-values (0.45, 0.53, and 0.45, respectively). At T0, ‘difficulty in walking’ scored 100 points, whereas ‘difficulty in maintaining balance’ and ‘unexplained weight loss’ scored 100 points at T5 and T9, respectively. In addition, the points belonging to ‘difficulty in walking’ decreased at T5 (84) and T9 (12). The items of social frailty had no points at T9. 

For the graphical presentation of the nomograms, we refer to Figure A1, Figure A2 and Figure A3.

### 3.3. Prediction of ADL Disability

Table 5 shows the points for the items in the nomograms and the performance of the underlying models (*Rsq*-values) over time in predicting the ADL disability scores (0.43, 0.52, and 0.41, respectively). At T0 and T5, the score for ‘difficulty in walking’ was 100 points, reducing to 28 points at T9. At T9, ‘unexplained weight loss’ scored 100 points. At T0 and T5, six items referring to psychological and social frailty scored points. However, at T9, only one item (feeling down) belonging to psychological frailty had points (25). We refer to Figure A4, Figure A5 and Figure A6 for the graphical presentation of the nomograms.

### 3.4. Prediction of IADL Disability

Table 6 shows the points for the items in the nomograms and the performance of the underlying models (*Rsq*-values) over time in predicting the IADL disability scores. The *Rsq*-values were 0.39, 0.44, and 0.44, respectively. At T0, ‘difficulty in walking’ scored 100 points; however, this item received no points at T9. ‘Unexplained weight loss’ scored 100 points at T5 and T9. It is remarkable that ‘lack of strength in the hands’ and ‘problems with memory’ scored much higher at T9 compared with both T0 and T5. Regarding the social frailty items, only ‘lack of social relations’ received points at T0. See Figure A7, Figure A8 and Figure A9 for the graphical presentation of the nomograms.

## 4. Discussion

Disability is associated with increased healthcare utilization [11,12], lower quality of life [10,13], and premature death [14,15,16] among older people. Frailty, which is also common in older people, can be considered a determinant of disability [18,19,20]. Thus, to prevent disability, it is useful to understand the contribution of individual frailty components in predicting disability. This insight is important for healthcare professionals to be able to intervene early so that disability is prevented or at least delayed. In this study, we aimed to develop nomograms with items of the Tilburg Frailty Indicator (TFI) [24] for predicting disability, assessed with the Groningen Activity Restriction Scale (GARS) [27], where we distinguish between total disability, ADL disability, and IADL disability. We used both cross-sectional data and longitudinal data, with a follow-up of five and nine years of a sample of Dutch community-dwelling people aged ≥75 years. Our premise was that the nomograms for predicting disability would differ by outcome variable and by time point. In this section, we will discuss only the main findings.

The nomograms derived from linear regression analyses showed the points that must be given to the 15 frailty components assessed with the TFI. The three monograms belonging to total, ADL, and IADL disability showed that the frailty item ‘difficulty in walking’ is the most important predictor at baseline (T0); ‘difficulty in walking’ scored 100 points for all three disability outcomes. The relative importance of this TFI item was less at T5 (total, IADL) and T9; regarding IADL disability, ‘difficulty in walking’ scored 0 points at T9. A systematic review including 28 studies showed that slow walking speed was one of the most powerful predictors of ADL disability in community-dwelling older people aged 65 years or older [20]. Another systematic review also found an association between walking speed and the probability of disability [32]. A recently developed prediction model demonstrated that age, walking speed, and cognitive function were the strongest predictors of disability-free survival in healthy older people [33]. We only found two studies, both longitudinal, carried out in the Netherlands aimed at predicting disability assessed with the GARS by frailty items [34,35]. The first used the physical subscale of the TFI [35]; this study was conducted in a sample of 429 Dutch people aged ≥ 65 years and showed, based on linear regression analyses, that slowness predicted both total and IADL using a follow-up of two and a half years. The second study aimed to predict ADL and IADL disability using an objective measure of walking speed, the Timed Up and Go (TUG) test, with a follow-up period of one year [34]. After controlling for previous disability and other predictors (background characteristics, body mass index, physical activity, handgrip strength, fatigue, balance), walking speed was predictive for total, ADL, and IADL disability.

Another physical frailty component, ‘unexplained weight loss,’ was the most important predictor for total, ADL, and IADL disability at T9; this item achieved the highest number of points for all three outcome variables (100 points). Unexplained weight loss is common in frail older people, with prevalence figures rising to 27% [36]. Based on this finding, prevention of unexplained weight loss in older people seems to be very important, especially since unexplained weight loss in this target group is associated with increased morbidity as well as increased mortality [37]. Additionally, after controlling for health, functional status, and social network, this frailty item is an important predictor of early institutionalization [36]. According to Alibhai et al. [38], for managing unexplained weight loss, identifying and treating the underlying causes (e.g., malignant disease, psychiatric disorder, gastrointestinal disease) should be the first priority.

Lack of strength in the hands and problems with memory were important predictors for IADL disability, in particular at T9, with 93 and 84 points, respectively. This was not the case with ADL disability; at T9, ‘lack of strength in the hands’ scored 36 points and ‘problems with memory’ scored 0 points. In addition to the two items mentioned earlier (difficulty in walking, unintentional weight loss), ‘lack of strength in the hands’ belongs to the five criteria of the phenotype of frailty by Fried et al. [22]. Our finding is confirmed by an umbrella review of systematic reviews using meta-analyses of observational studies [39]. In this review, handgrip strength was not only considered a useful indicator for disability but also for general health status and mortality. It should be noted, however, that the operational definition of disability was different from our definition, IADL disability assessed with the GARS. With regard to the frailty item ‘problems with memory’, referring to cognitive impairment, many previous studies showed that cognition predicts disability, e.g., Shimada et al. [40] and St John et al. [41]. In a community-dwelling baseline sample of 1715 people aged ≥65 years, cognition determined with the mini-mental state examination (MMSE) predicted disability 5 years later [41]. In addition, in a Japanese sample consisting of 4290 community-dwelling older people aged 65 years or older, cognitive impairment showed a significant association with disability [40].

The TFI assesses physical, psychological, and social frailty. The findings of the present study showed that the eight items belonging to physical frailty predicted total, ADL and IADL disability to a greater extent, in particular, difficulty in walking and unintentional weight loss, or, to a lesser extent, especially poor hearing, poor vision, and physical tiredness. The items referring to psychological and social frailty received far fewer points. The social frailty item ‘lack of social support’ was the only frailty item that scored zero points. For predicting disability, psychological and social frailty are obviously less important than physical frailty. It is assumed that in other adverse outcomes of frailty (e.g., increased healthcare utilization, lower quality of life), the nomograms will strongly differ, and more points will be given to psychological and social frailty items. If we consider the prediction of ADL and IADL disability using the TFI, then we can observe that the *Rsq*-values were nearly the same; the main difference in *Rsq*-values existed between ADL and IADL at T5 (0.52 versus 0.44). In addition, for both T0 and T5, there were more predictors of disability than for T9. At T9, there were only nine, seven, and eight predictors for total, ADL, and IADL disability, respectively. This finding is not so surprising; after all, making a long-term prediction is trickier than making a short-term prediction.

Our study has several limitations. Firstly, the generalisability of the findings may be questioned because the response rate at baseline was only 42%. Secondly, for the prediction of total, ADL, and IADL disability, we only used frailty items and did not control for background characteristics of the participants, e.g., individual diseases or multimorbidity and age. However, in a previous Dutch study using the TFI and GARS, it was observed after sequential linear regression analyses that only previous disability and age significantly contributed to the prediction of disability; multimorbidity and other background characteristics (e.g., sex, education, income) did not [35]. Third, the sample size can be considered small at T9. Because of the high age at baseline (mean 80.3 years; SD 3.8), many older people were unable to participate in the follow-up of this study, especially for the measurement of disability at T9. In a previous study using the same sample, it was observed that 162 died between 2008 and 2015 [42]. Finally, we would like to indicate that discussion of our findings in light of findings from previous studies has its limitations because both frailty and disability were usually measured with instruments other than the TFI and the GARS, such as the phenotype of frailty [22], the FI [23] for assessing frailty, and the Katz Scale [43] and the Lawton and Brody Scale [44] for assessing ADL and IADL disability, respectively.

## 5. Conclusions

In the present study, we developed nomograms for the prediction of total, ADL, and IADL disability based on physical, psychological, and social items of the Tilburg Frailty Indicator (TFI). We showed that the frailty items scored different points, especially over time. In other words, not every item was equally important in predicting disability. In particular, the frailty items ‘difficulty in walking’ and ‘unexplained weight loss’ were important predictors of disability. Healthcare professionals need to focus on these two items in order to prevent disability and to prevent the worsening of disability. Additionally, we can conclude that the points given to frailty items differed between total, ADL, and IADL disability and also differed with regard to years of follow-up. Creating one monogram that does justice to this seems impossible.

## Figures and Tables

**Table 1 healthcare-11-01150-t001:** Characteristics of the participants (T0, *n* = 479; T5, *n* = 160; T9, *n* = 77).

Characteristics	T0		T5		T9	
	*n*	%	*n*	%	*n*	%
**Gender**						
Man	207	43.2	76	47.5	37	48.1
Woman	272	56.8	84	52.5	40	51.9
**Marital status**						
Married or cohabiting	238	49.8	71	44.4	30	39.0
Not married	45	9.4	12	7.5	6	7.8
Divorced	15	3.1	9	5.6	5	6.5
Widowed	180	37.7	68	42.5	36	46.8
**Country of birth**						
The Netherlands	461	96.6	159	99.4	75	97.4
Other	16	3.4	1	0.6	2	2.6
**Educational level**						
No or primary	181	38.1	51	31.9	20	26.0
Secondary education	221	46.5	75	46.9	40	51.9
Higher education	73	15.4	34	21.2	17	22.1
**Monthly income in Euro ***						
Less than 600	12	2.7	2	1.3	2	2.7
601–900	71	16.2	6	4.0	3	4.1
901–1200	106	24.2	33	22.1	8	11.0
1201–1500	57	13.0	21	14.1	14	19.2
1501–1800	67	15.3	23	15.4	13	17.8
1801–2100	48	11.0	25	16.8	8	11.0
2101 or more	77	17.6	39	26.2	25	34.2
**Continuous variables (mean, sd)**						
Age	80.3	3.8	84.0	3.2	87.4	2.9
Frailty score	4.7	3.0	5.0	3.0	5.8	3.0
Disability total	26.7	9.6	28.7	10.1	33.3	12.5
ADL disability	14.6	4.8	15.6	5.5	17.8	6.8
IADL disability	12.2	5.4	13.1	5.5	15.5	6.6

* Missing values at T0 = 41, at T5 = 11, at T9 = 4.

**Table 2 healthcare-11-01150-t002:** *p*-values for changes in distribution categorical variables.

	T0–T5			T5–T9			T0–T9		
	Chi-square	df	*p*-value	Chi-square	df	*p*-value	Chi-square	df	*p*-value
Gender	1:197	1	0:274	0:009	1	0:923	0:734	1	0:392
Marital status	5:715	3	0:126	0:948	3	0:814	6:484	3	0:090
Country of birth	3:677	1	0:055	4:823	1	0:028	0:136	1	0:712
Educational level	5:235	2	0:073	1:289	2	0:525	5:716	2	0:057
Monthly income in euro	25:966	6	0:000	10:085	6	0:121	25:850	6	0:000

**Table 3 healthcare-11-01150-t003:** Characteristics of the 15 TFI items.

	Total	
	*n*	%
Physically unhealthy		
no	334	70.8
yes	138	29.2
Unexplained weight loss		
no	442	92.5
yes	36	7.5
Difficulty in walking		
no	248	51.9
yes	230	48.1
Difficulty in maintaining balance		
no	308	65.0
yes	166	35.0
Poor hearing		
no	301	63.4
yes	174	36.6
Poor vision		
no	372	78.6
yes	101	21.4
Lack of strength in the hands		
no	315	65.8
yes	164	34.2
Physical tiredness		
no	261	54.6
yes	217	45.4
Problems with memory		
no	432	90.4
yes	46	9.6
Feeling down		
no	286	59.8
yes	192	40.2
Feeling nervous or anxious		
no	330	69.0
yes	148	31.0
Unable to cope with problems		
no	401	85.0
yes	71	15.0
Living alone		
no	250	52.2
yes	229	47.8
Lack of social relations		
no	196	41.0
yes	282	59.0
Lack of social support		
no	398	83.6
yes	78	16.4

**Table 4 healthcare-11-01150-t004:** *Rsq*-values and relative importance TFI items for the total disability scores.

	T0	T5	T9
Performance model			
*Rsq*	0.45	0.53	0.45
Predictors			
Physically unhealthy	57	25	36
Unexplained weight loss	56	56	100
Difficulty in walking	100	84	12
Difficulty in maintaining balance	47	100	25
Poor hearing	3	60	9
Poor vision	14	24	19
Lack of strength in the hands	35	77	60
Physical tiredness	37	52	0
Problems with memory	35	7	18
Feeling down	0	9	12
Feeling nervous or anxious	28	6	0
Unable to cope with problems	30	34	0
Living alone	0	0	0
Lack of social relations	19	13	0
Lack of social support	0	0	0

**Table 5 healthcare-11-01150-t005:** *Rsq*-values and relative importance TFI items for the ADL disability scores.

	T0	T5	T9
Performance model			
*Rsq*	0.43	0.52	0.41
Predictors			
Physically unhealthy	70	11	34
Unexplained weight loss	79	0	100
Difficulty in walking	100	100	28
Difficulty in maintaining balance	49	97	32
Poor hearing	0	50	0
Poor vision	4	11	5
Lack of strength in the hands	36	70	36
Physical tiredness	37	34	0
Problems with memory	5	0	0
Feeling down	0	0	25
Feeling nervous or anxious	30	10	0
Unable to cope with problems	10	42	0
Living alone	22	14	0
Lack of social relations	11	29	0
Lack of social support	0	0	0

**Table 6 healthcare-11-01150-t006:** *Rsq*-values and relative importance TFI items for the IADL disability scores.

	T0	T5	T9
Performance model			
*Rsq*	0.39	0.44	0.44
Predictors			
Physically unhealthy	41	21	39
Unexplained weight loss	46	100	100
Difficulty in walking	100	31	0
Difficulty in maintaining balance	44	51	16
Poor hearing	8	36	22
Poor vision	22	20	38
Lack of strength in the hands	47	43	93
Physical tiredness	42	37	0
Problems with memory	58	28	84
Feeling down	0	11	0
Feeling nervous or anxious	29	1	1
Unable to cope with problems	43	12	0
Living alone	0	0	0
Lack of social relations	21	0	0
Lack of social support	0	0	0

## Data Availability

The data presented in this study are available on request from the corresponding author.

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
