# Peer review of "Predicting Disability Using a Nomogram of the Tilburg Frailty Indicator (TFI)"

_healthcare, 2023, doi:10.3390/healthcare11081150_

Round 1
Reviewer 1 Report
Thank you for the opportunity to review this manuscript developing nomograms with items of the Tilburg Frailty Indicator (TFI) as predictors by using cross-sectional and longitudinal data, focusing on the prediction of total disability, disability in activities of daily living (ADL) and disability in instrumental activities of daily living (IADL). This study showed that the TFI items scored different points, especially over, and “Difficulty in walking” and “unexplained weight loss” appeared to be important predictors of disability. This reviewer has certain questions, which should be addressed by authors prior to publication.
It might be noteworthy that some variables of frailty predicted disability, and predicting disability differed between follow-up periods. However, it seems that the sample size was too small to analyze the nomograms using liner regression models.
This study investigated to predict disability using TFI at each study period (T0, T5, and T9). Could any baseline variables of TFI predict disability at follow-up periods (T5 and T9)? It would be better to add explanation why you had focused on each study period rather than longitudinal changes.
There is a lack of information for study population. Were there any exclusion criteria such as severe cognitive impairment, people who admitted a nursing home? How many people died during follow-up period? The reasons why they did not answer the questionnaires at follow-up period might be helpful.
Reviewer 2 Report
Dear,
The authors have conducted an observational to study to develop nomograms with items of the Tilburg Frailty Indicator (TFI) as predictors by using cross-sectional and longitudinal data (follow-up of five and nine years), focusing on the prediction of total disability, disability in activities of daily living (ADL) and disability in instrumental activities of daily living (IADL).
Please give your statement to the following points:
1. Abstract
- No problem
2. Introduction
- Please specify the aim of the study and clinical message that the authors want to send
3. Materials and Methods
Please specify if they were followed the STROBE guidelines and IRB authorization
Inclusion criteria and Exclusion criteria: please, better specify.
Sample size: please, better specify
power and calculation and statistical plan: well explained.
Please better specify if there are missing data
4. Results
Complete
5.Discussion
- please specify the aim of the study and clinical message that the authors want to send
6. Tables and Figures
- no problem
7. References
Please check the journal’s guidelines
Best regards
Reviewer 3 Report
Abstract
More on the content in the final conclusion rather than on the method
Introduction
Page 1, line 31
The definition of disability reported by the authors can be enriched by more recent references. In particular, they should refer to the international organizations’ documents where the definitions are provided.
Page 2, line74
The authors set the level of questions to the items of the TFI while the core question should be the multiple dimensions under the attention The tool should be seen just as the means to address the question.
Page 3, line 100.
The disability-appropriate language should propose “persons with disability” instead of “disabled” to put the person at the center of attention.
Line 107.
Authors should also for this questionnaire keep attention to the dimensions addressed and consider the construct, not the role (predictor variable is appropriate when they describe the analysis).
More details on the choice of nomograms can help readers understand the choices of the authors.
Page 4, line 156.
Please give the readers a reading of the results obtained from the analysis.
Page 5.
The graphic organization of the tables makes it more difficult to explore them. If possible, please reduce the size of the font. It could be also useful to make the heading of the table on this page more explicit.
Discussion
Across page 9 authors often refer to previous studies to support the study findings, the same studies for which they suggest being cautious. It is important then to describe more in detail the points that make the comparison relevant showing the commonalities in the tools and the level of analysis that make that comparison suitable.
Line 269.
The comments are “tricky.”
The conclusions
Page 10, line 293
Given the comments made on page 9, the results are relevant not only to preventing disability. They might be relevant also to prevent the worsening of disability.
Based on the results authors may also anticipate the direction and potential relevance of continuing the longitudinal analysis.
Round 2
Reviewer 1 Report
I do believe that the authors have addressed the issues that I raised and that the manuscript is significantly better than before.